# Effect of Preheating Temperature on Thermal–Mechanical Properties of Dry Vibrating MgO-Based Material Lining in the Tundish

**DOI:** 10.3390/ma15217699

**Published:** 2022-11-02

**Authors:** Xiaodong Deng, Jianli Li, Xiao Xie

**Affiliations:** 1The State Key Laboratory of Refractories and Metallurgy, Wuhan University of Science and Technology, Wuhan 430081, China; 2Hubei Provincial Key Laboratory for New Processes of Ironmaking and Steelmaking, Wuhan University of Science and Technology, Wuhan 430081, China; 3School of Computer Science and Technology, Wuhan University of Science and Technology, Wuhan 430081, China

**Keywords:** dry vibrating material, collapsed ladle, mathematical modeling, temperature field, stress field, strain field

## Abstract

For the collapse of the working layer of dry vibrating material during preheating, the four-strand tundish of a steel plant was taken as a prototype for numerical simulation. The software ANSYS was used to calculate the temperature field and stress and strain field on the working layer under three preheating stages through the indirect coupling method. The results show that during the preheating process, the temperature field distribution on the hot surface of the working layer gradually develops toward uniformity with the increase in preheating temperature. However, the temperature gradient between the cold and hot surfaces increases subsequently, and the highest temperature between the cold and hot surfaces reaches 145.31 °C in the big fire stage. The stress on the top of the working layer is much larger than in other areas, and the maximum tensile stress on the top reaches 39.06 MPa in the third stage of preheating. Therefore, the damage to the working layer starts from the top of the tundish. In addition, the strain of the area near the sidewall burner nozzle in the casting area is much larger than that in the middle burner area with the increase in preheating temperature. Thus, the working layer near the sidewall burner nozzle is more prone to damage and collapse compared with the middle burner nozzle.

## 1. Introduction

Continuous casting tundish, working as a buffer and distributor of liquid steel between the ladle and continuous casting molds, plays a key role in affecting the performance of casting and solidification, as well as the quality of final products [1,2,3]. Therefore, the quality of the refractory material of the tundish is important for the full development and improvement of the metallurgical performance of the tundish. The development of the tundish working layer can be divided into four stages: (i) no working layer stage, (ii) insulation board stage, (iii) gunning material stage, and (iv) dry vibrating material stage [4]. Generally, tundish refractory lining mainly includes an insulation layer, a permanent layer, and a working layer. The working layer is in contact with molten steel. Thus, the working lining material should have good slag resistance, thermal shock resistance, volume stability, excellent thermal insulation performance, and high-temperature strength [5,6,7]. The dry vibrating material as the fourth generation of tundish working layer material, not only has the advantages of both insulation board and gunning material but also has the advantages of easy construction, high thermal efficiency, fast tundish turnover, long service life, and low energy consumption, which is widely used in the metallurgical industry of continuous casting tundish [8,9].

With the development of continuous casting technology, it is gradually realized that the thermal state of the tundish has an important role in controlling the temperature of the molten steel in the tundish, maintaining the stability of the superheat of the molten steel, improving the quality of the continuous casting billet, saving energy, and prolonging the service life of the tundish [10,11,12]. During the continuous casting process, the molten steel will be poured from the ladle into the tundish. There is conductive heat loss through the wall of the tundish and radiative heat loss through the bath surface [13,14]. In order to reduce the heat loss of the molten steel, the lining can be preheated to absorb a large amount of heat before pouring to reduce the temperature difference between the lining and the molten steel.

However, during the use of the dry vibrating material working layer, it was found that the dry vibrating material with phenolic resin as the low-temperature binding agent will undergo an oxidation reaction to form a decarbonized layer during the preheating process, and the original physical properties of the working layer of the tundish will be changed by the formation of the decarbonized layer [15]. When the preheating regime of the tundish is imperfect, the working layer is prone to large area collapse and local spalling, which leads to the production of instable running. In order to solve this problem, Shi et al. [16] analyzed the current status of refractory temperature and thermal stress research, and established and analyzed the temperature and thermal stress distribution of two types of refractories for ladles using APDL language. Li et al. [17] carried out a large number of follow-up analyses on the collapsed tundish of magnesium dry vibrating material tundish and optimized and improved the preheating process by testing and analyzing the strength of the dry vibrating material preheating process and the change law of the preheating temperature rise.

In the present study, a 1:1 model with a four-strand tundish of a steel mill as the prototype was established. Finite element simulation of the temperature field and the stress field during preheating in a tundish body was carried out by ANSYS software. According to the distribution and changes in temperature, stress, and strain fields on the working layer during the preheating process, the most vulnerable areas of the working layer were analyzed. This research aims to identify the stress distribution at the critical point and analyze the cause of the damage. The obtained results are of great significance to the normal operation of the continuous casting process.

## 2. Model Description

### 2.1. Fluid Flow and Heat Transfer

In the preheating process of tundish, the mixture of gas and air enters the tundish after ignition. Then, the combusted gas transfers heat to the tundish lining through convection and radiation in the flow process. During the preheating process fluid, flow, and heat transfer are modeled using conservation equations of mass Equation (1), momentum Equation (2), and energy Equation (3):(1)$$\frac{\partial \rho }{\partial t}+\frac{\partial \left(\rho u_{j}\right)}{\partial x_{j}}=0$$
(2)$$\rho \frac{\partial u_{i}}{\partial t}+\rho u_{j}\frac{\partial u_{i}}{\partial x_{i}}=-\frac{\partial P}{\partial x_{i}}+\frac{\partial }{\partial x_{j}}\left[\left(\mu +\mu_{t}\right)\left\{\frac{\partial u_{i}}{\partial x_{j}}+\frac{\partial u_{j}}{\partial x_{i}}\right\}\right]+g_{i}\left(\rho -\rho_{0}\right)$$
(3)$$\rho C_{p}\frac{\partial T}{\partial t}+\rho C_{p}\frac{\partial \left(u_{j}T\right)}{\partial x_{j}}=\frac{\partial }{\partial x_{j}}\left[\left(k_{0}+\frac{C_{p}\mu_{t}}{Pr_{t}}\right)\frac{\partial T}{\partial x_{j}}\right]+S_{T}$$
where *ρ* is the density; *C_P_* is the heat capacity; *μ_t_* is the turbulent viscosity; *Pr_t_* is the turbulent Prandtl number; *S_T_* is the source term of the energy equation; *u* is the velocity; *x* is the Cartesian space coordinates, and subscripts *i*, *j* are for the coordinate directions.

The standard *k*-*ε* models [18] are shown in Equations (4) and (5):(4)$$\frac{\partial }{\partial t}\left(\rho k\right)+\frac{\partial }{\partial x_{i}}\left(\rho ku_{i}\right)=\frac{\partial }{\partial x_{j}}\left[\left(\mu +\frac{\mu_{t}}{\sigma_{k}}\right)\frac{\partial k}{\partial x_{j}}\right]+G_{k}+G_{b}-\rho \varepsilon -\gamma_{M}+S_{K}$$
(5)$$\frac{\partial }{\partial t}\left(\rho \varepsilon \right)+\frac{\partial }{\partial x_{i}}\left(\rho \varepsilon u_{i}\right)=\frac{\partial }{\partial x_{j}}\left[\left(\mu +\frac{\mu_{t}}{\sigma_{\varepsilon }}\right)\frac{\partial k}{\partial x_{j}}\right]+C_{1}\frac{\varepsilon }{k}\left(G_{k}+C_{3}G_{b}\right)-C_{2}\rho \frac{\varepsilon^{2}}{k}+S_{\varepsilon }$$
where *k* is the kinetic energy of turbulence per unit mass; *ε_F_* is the turbulent energy dissipation rate; *μ* is the molecular viscosity; *μ_t_* is the turbulent viscosity; *G_k_* is the turbulent kinetic energy generated by the laminar velocity gradient; *G_b_* is the turbulent kinetic energy generated by buoyancy [19]; *γ_M_* is the fluctuation generated by the transition diffusion in compressible turbulence; *σ_k_* and *σ_ε_* are the turbulent Prandtl numbers of *k* and *ε*, respectively; *C*_1_, *C*_2_, and *C*_3_ are the constants; and *S_K_* and *S_ε_* are the source term of the turbulent kinetic energy(*k*) and its dissipation rate (*ε_F_*).

As mentioned by Launder and Spalding [18], the other values for model constants in this study were *C*_1_ = 1.44, *C*_2_ = 1.92, *σ_k_* = 1.0, and *σ_ε_* = 1.3. For buoyant shear layers for which the main flow direction is aligned with the direction of gravity, *C*_3_ becomes 1. For buoyant shear layers that are perpendicular to the gravitational vector, *C*_3_ becomes 0.

### 2.2. Solid Heat Transfer and Thermal Stress

During the preheating process, the heat transfer of the tundish package can be considered as a three-dimensional steady-state heat transfer, and the heat-transfer behavior is expressed by the following Equation (6):(6)$$\frac{\partial^{2}T}{\partial x}+\frac{\partial^{2}T}{\partial y}+\frac{\partial^{2}T}{\partial z}=0$$
where *T* is the temperature; *x*, *y*, and *z* are the x, y, and z coordinate directions.

According to the temperature distribution and thermal expansive coefficient of each part of the tundish lining, the deformation is calculated in the special constraints. Then, the strain of each point of the tundish is calculated using the deformation of tundish displacement with the geometric equation. Finally, the stress in the tundish of each point is calculated through the strain according to the physical equation of tundish material [20,21].

The thermal stress field geometry equation is used to calculate the relationship between strain and displacement. It can be expressed by the following Equation (7).
(7)$$\varepsilon =\left[\begin{array}{ccc} \frac{\partial }{\partial x} & 0 & 0\\ 0 & \frac{\partial }{\partial y} & 0\\ 0 & 0 & \frac{\partial }{\partial z}\\ \frac{\partial }{\partial y} & \frac{\partial }{\partial x} & 0\\ 0 & \frac{\partial }{\partial z} & \frac{\partial }{\partial y}\\ \frac{\partial }{\partial z} & 0 & \frac{\partial }{\partial x} \end{array}\right]\lambda$$
where $\varepsilon ={\left[\begin{array}{cccccc} \varepsilon_{x} & \varepsilon_{y} & \varepsilon_{z} & \gamma_{xy} & \gamma_{xz} & \gamma_{yz} \end{array}\right]}^{T}$ is the strain at any point in tundish; $\lambda ={\left[\begin{array}{ccc} u & v & w \end{array}\right]}^{T}$ is the displacement along the directions of *x*, *y*, and *z*.

The physics equation of the stress field is used to calculate the relationship between strain and stress. It can be expressed by the following Equation (8).
(8)$$\sigma =\frac{E\left(1-v\right)}{\left(1+v\right)\left(1+2v\right)}\left[\begin{array}{cccccc} 1 & \frac{v}{1-v} & \frac{v}{1-v} & 0 & 0 & 0\\ \frac{v}{1-v} & 1 & \frac{v}{1-v} & 0 & 0 & 0\\ \frac{v}{1-v} & \frac{v}{1-v} & 1 & 0 & 0 & 0\\ 0 & 0 & 0 & \frac{1-2v}{2\left(1-v\right)} & 0 & 0\\ 0 & 0 & 0 & 0 & \frac{1-2v}{2\left(1-v\right)} & 0\\ 0 & 0 & 0 & 0 & 0 & \frac{1-2v}{2\left(1-v\right)} \end{array}\right]$$
where *E* is the elastic modulus; *v* is Poisson’s radio; *σ* is the stress; and *ε* is the strain.

According to the above stress–strain relations, any stress of each point is calculated by each point of its inner strain obtained from the previous step, and the products are the object of the force, with the force meeting the balance equation.

### 2.3. Materials and Methods

In the present study, the characterization of the working and permanent layers required for the simulation calculations was made in the laboratory. The raw materials of the working and permanent layers were taken from the industrial raw materials of a steel mill. Test samples of the dry vibrating material were put into a mold with an inner size of 40 mm × 40 mm × 160 mm, followed by tamping until the density of raw materials was about 2.40 g/cm^3^. These prepared samples were initially dried at 220 °C for 3 h to obtain handling strength, and then were fired in an electric furnace at 1000 °C for 2 h under air conditions (oxidizing atmosphere) and buried carbon conditions (reducing atmosphere), respectively, where the materials were heated at a rate of 5 °C/min, followed by cooling to room temperature in the furnace. Test samples of the permanent layer materials were put into a mold with the inner size of 40 mm × 40 mm × 160 mm, followed by vibrating. After heating for 3 h at 220 °C condition, all the specimens were fired at 1000 °C for 2 h in an electric furnace under an air atmosphere condition with the heated rate of 5 °C/min and then cooled down to room temperature. The bulk density of three test samples was evaluated using Archimedes’ principle in kerosene medium according to GB/T 2997-2015 standard.

Thermal conductivity of the disc specimens with Φ180 mm × 20 mm at 350 °C, 600 °C, 800 °C, and 1000 °C was evaluated using the hot-wire method per the standard GB/T 5990-2006. The characterization of the dynamic elastic modulus at room temperature was evaluated according to GB/T 30758-2014. The cylindrical specimens of Φ8 mm × 50 mm were prepared, and the thermal expansion rate of the specimen was measured at 100–1400 °C using a high-temperature thermal (Precondar, Luoyang, China) expansion meter per standard GB/T 7320-2008.

### 2.4. Geometry, Mesh, and Boundary Conditions

In this study, a four-strand tundish of a steel mill is modeled as a prototype 1:1. Considering the symmetry of the tundish structure, half of the model is taken as the calculation area to reduce the calculation volume. The carbon element in the dry vibrating material of the working layer will react with the oxidizing substances in the preheating gas during the preheating process of the tundish, resulting in the delamination of the working layer and the generation of two layers of decarbonized layer and the original layer. Therefore, the working tundish lining with two layers was modeled, and the decarburization layer thickness was 25% of the whole working layer. The specific geometric model is shown in Figure 1a. The tundish body is composed of the original layer, decarbonized layer, permanent layer, insulation layer, steel shell, slag retaining wall, flow control device, and lid. The burner nozzle in the center of the pouring area on the lid is burner No. 1 and the burner nozzle in the pouring area near the side wall is burner No. 2. In addition, the burner nozzle in the impact area is burner No. 3. The physical properties of refractory linings used in this simulation are shown in Table 1 and Table 2. The characterization of the insulation layer and the steel shell in Table 1 can be found by Zhang [22].

The model is meshed using ANSYS MESH software. Based on the hybrid mesh division method, the areas of different complexity are divided into meshes of different sizes and densities to reduce the number of meshes as much as possible while ensuring calculation accuracy, so as to save the calculation resources and calculation time. The total number of meshes in the tundish model is about 8 million, in which the meshes of the working layer and burner of the tundish are encrypted. The mesh division results are shown in Figure 1b.

Non-slip conditions were applied on all wall boundaries for the fluid phase. A constant velocity flow was used at the inlet. At the outlet of the tundish, the outflow was applied. The heat loss is calculated based on the heat transfer coefficient at the side and bottom walls and the environment temperature. A summary of input parameters and boundary conditions used for computational fluid dynamics simulations is provided in Table 3. The heating process of the tundish is composed of three stages of preheating, namely, the first stage of preheating, the second stage, and the third stage of preheating. The preheating parameters for each stage are shown in Table 4. The data in Table 4 are from the steel mill site.

## 3. Results

### 3.1. Temperature Field Distribution

Figure 2 shows the temperature field distribution of the hot surface of the wall in lengthwise direction of the working layer. The result shows that with the increase in preheating temperature, the temperature distribution on the hot surface of the working layer gradually tends to be uniform. As shown in Figure 2a,b, the temperature field distribution on the hot surface of the working layer is less uniform at the first and second stages of preheating. The red area is mainly concentrated in the area of the No. 1 burner nozzle and the bottom area of the No. 2 burner nozzle. The distribution area is small, and the temperature difference between zones is 16 °C and 20 °C. Figure 2c displays that the distribution of the temperature field of the hot surface tends to be uniform during the third stage of preheating, and the temperature difference between the red zones is only 11.6 °C.

Figure 3 shows the temperature field distribution of the cold surface. The result shows that the trend of temperature field change on the cold side of the working layer is slightly different from that on the hot side. The temperature field on the cold side does not develop toward uniformity with the increase in preheating temperature. The high-temperature area is mainly concentrated in the middle and lower regions of the wall, and in the region near the top of the tundish, the working layer temperature is lower. There is a larger temperature gradient compared with the middle and lower regions. In addition, the temperature on the cold side has a large temperature difference compared with the temperature on the hot side. The temperature gradient between the cold and hot sides further expands with the increase in preheating temperature. In the third stage of preheating, the difference between the highest temperature on the cold side in Figure 3c and the highest temperature on the hot side in Figure 2c is 145.31 °C, and the temperature gradient further increases in the orange temperature zone of both figures. The results of this calculation indicate that there is a large temperature gradient in the longitudinal direction in the working layer of the dry vibrating material during the preheating process, which increases the possibility of thermal shock damage to the material and is harmful to the pouring of the tundish [20].

### 3.2. Stress and Strain Field Distribution

Figure 4 shows the distribution of the maximum principal stress applied to the hot surface of the wall in lengthwise direction at different preheating stages, which is oriented perpendicular to the wall of the working layer. When the maximum principal stress is positive, the stress is tensile stress and the direction is toward the fluid domain; when the maximum principal stress is negative, the stress is compressive stress and the direction is toward the permanent layer. As shown in Figure 4, the boundary region of the working layer belongs to the stress concentration region and the stress increases significantly with the increase in the preheating temperature. However, the maximum principal stress suffered by the working layer in the middle region has less change in stress value with the increase in the preheating temperature. The tensile stress at the top of the wall in the boundary region is significantly greater than that at the sides and bottom, which indicates that the possibility of cracks from the top of the working layer during preheating is much greater than that in other regions.

Figure 5 shows the maximum principal strain field on the cold surface of the working layer. The direction of the strain field is consistent with the direction of the maximum principal stress. As shown in Figure 5, the area where positive deformation occurred in the working layer was mainly concentrated in the middle of the wall, while the area where negative deformation occurred was distributed in the boundary area of the working layer. This indicates that during the preheating process, the working layer material deformed toward the fluid domain in the middle area, while the boundary area deformed toward the permanent layer, which exacerbated the stress concentration in the boundary area to a certain extent. The maximum positive deformation area on the cold surface of the working layer gradually shifts from the bottom to the side with the increase in preheating temperature. In the second and third stages of preheating, the maximum positive deformation area was concentrated in the area near the side wall. Moreover, the larger the value of positive deformation, the greater the possibility of detachment between the working layer and the permanent layer. When the deformation volume reaches a certain value, the working layer will be separated from the permanent layer, which is harmful to the pouring of the tundish.

## 4. Analysis and Discussion

In order to judge more intuitively the stress changes suffered by the working layer, combined with the simulation results of the temperature field and stress field of the dry vibrating material of the tundish in the third stage of preheating, a straight line called path 1 and path 2 was made in the area corresponding to the center of the burner nozzles 1 and 2 of the working layer, respectively, to calculate the temperature field and stress field distribution on the paths. The two paths are located on the hot surface of the working layer, as shown in Figure 6, while the distribution curves of temperature and maximum principal stress on the paths are shown in Figure 7 and Figure 8. The horizontal coordinate 0 mm is the bottom of the middle pack, and L is the length on the path, and the larger its value, the closer the path is to the top of the pack. The red curve is the variation curve of the maximum principal stress with the path, and the black curve is the variation curve of temperature with the path.

As shown in Figure 7, the magnitude of the maximum principal stress in the region of 80–980 mm on path 1 varies extremely, and its stress value is close to 0 MPa, which cannot cause damage to the working layer. While in the bottom region of path 1, the maximum tensile stress is 1.336 MPa and the maximum compressive stress is 7.2 MPa. However, in the top region, the maximum tensile stress on path 1 is 3.03 MPa and the maximum compressive stress reaches 57.98 MPa. Figure 9 shows the strength characteristics of the dry vibrating material. As can be seen from the figure, the resin in the dry vibrating material is gradually cured with the increase in preheating temperature so that the working layer has a certain strength at low temperatures. However, the curing resin starts to oxidize and decompose at 200 °C, resulting in a loss in the strength of the working layer. After 1000 °C, the strength of the working layer gradually increases with the sintering of the dry vibrating material [23]. According to the temperature curve in Figure 7, it is known that the temperature at the top of the working layer is lower than in other areas. Thus, in the case where the dry vibrating material has lower strength and greater stress, the top of the working layer is more prone to be damaged, cracks are formed, and spalling occurs [17].

Figure 8 shows that the maximum tensile stress is 1.14 MPa and the maximum compressive stress is 9.50 MPa in the bottom area of path 2, while the maximum tensile stress is 5.34 MPa and the maximum compressive stress is 64.6 MPa in the top area of path 2. Compared with path 1, the stress in the top area of path 2 has increased significantly, while the temperature has decreased slightly. Combined with the strength characteristics of the dry vibrating material in Figure 9, it is known that the working layer is more prone to be damaged and cracked at path 2. On the other hand, according to Figure 5c, there is a possibility of separation of the working layer from the permanent layer. When cracks are generated at the top of path 2 and the cracks extend to the bottom of the working layer, the steel will penetrate the working layer along the cracks during the pouring process, which will lead to large spalling and tundish collapse.

## 5. Conclusions

(1)The temperature field distribution on the hot surface of the working layer gradually develops toward uniformity with the increase in preheating temperature, while the temperature gradient between the cold and hot surfaces increases. The highest temperature difference between the cold and hot surfaces reaches 201.31 °C during the third stage of preheating, which increases the possibility of thermal shock damage to the refractory of the working layer.(2)During the preheating process, the stress on the working layer is mainly concentrated in the boundary area. The maximum tensile stress at the top of the working layer reaches 39.06 MPa during the third stage of preheating, which is significantly greater than that in other areas. It can be seen that the possibility of damage from the top of the working layer to forming cracks is much greater than in other areas.(3)In the high-temperature preheating stage, the maximum principal strain in the area of the No. 2 burner nozzle is 0.0076198 mm, which is much larger than in other areas. When cracks are generated at the top of the working layer and the cracks extend downward, the working layer will collapse and spall in a large area due to the penetration of steel.

## Figures and Tables

**Figure 1 materials-15-07699-f001:**
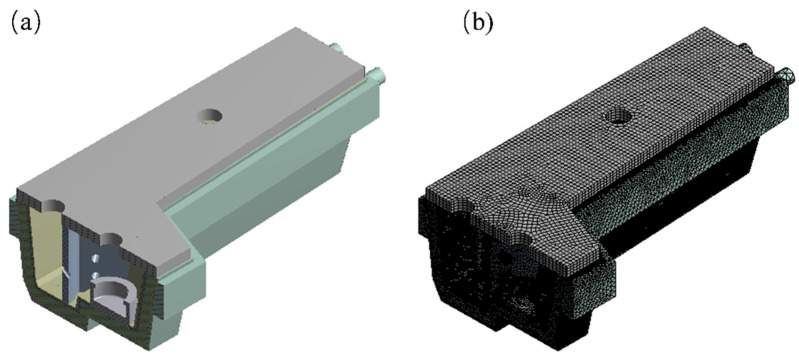
Geometry and mesh: (**a**) geometry; (**b**) grid of computational domain.

**Figure 2 materials-15-07699-f002:**
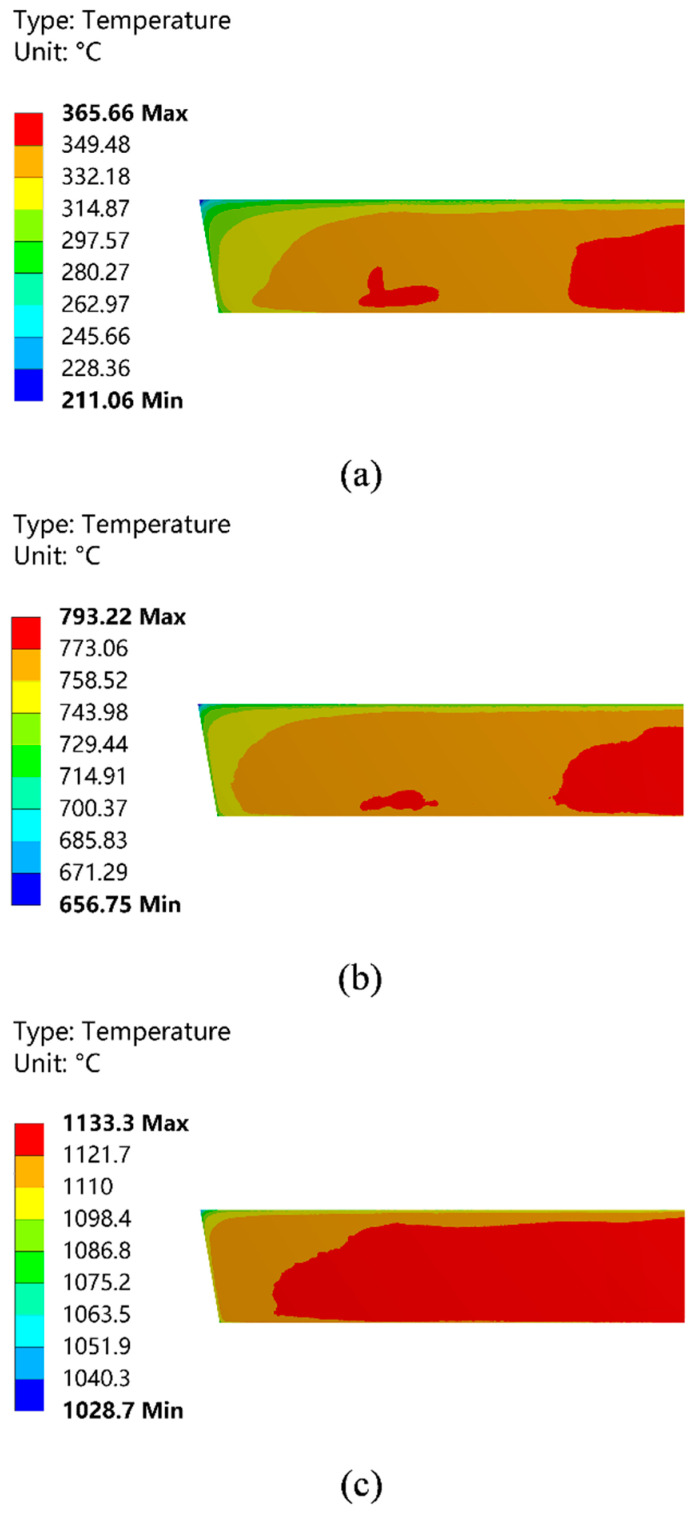
Temperature field distribution of the hot surface of the working layer. (**a**) The first stage of preheating; (**b**) the second stage of preheating; and (**c**) the third stage of preheating.

**Figure 3 materials-15-07699-f003:**
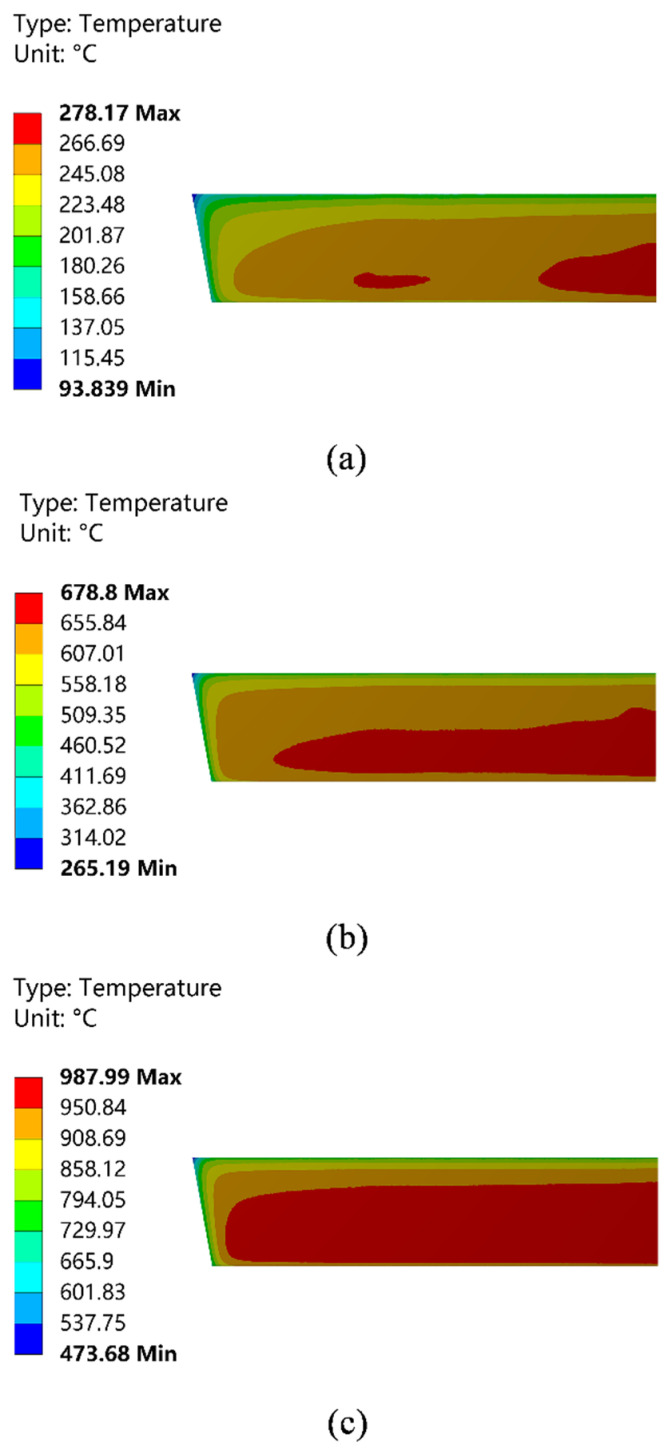
Temperature field distribution of the cold surface of the working layer. (**a**) The first stage of preheating; (**b**) the second stage of preheating; and (**c**) the third stage of preheating.

**Figure 4 materials-15-07699-f004:**
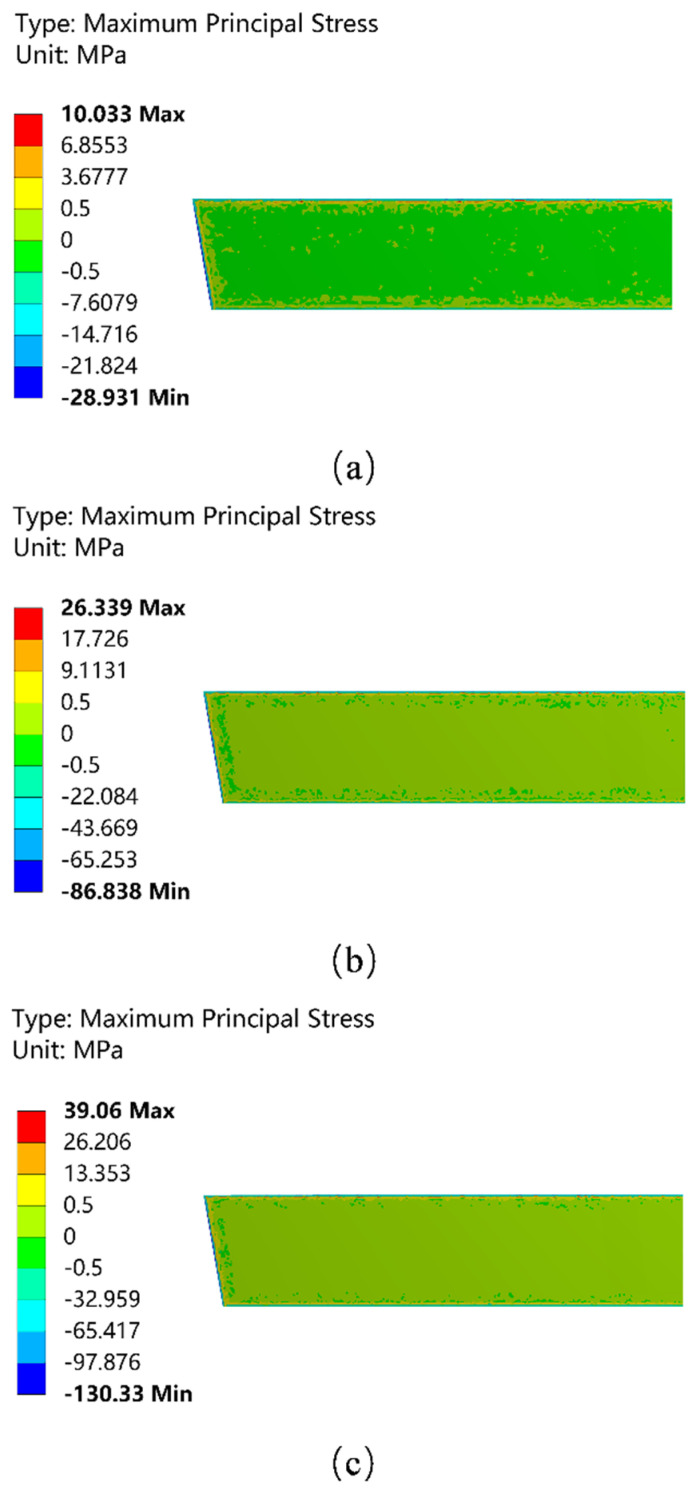
Distribution of the maximum principal equivalent stress field of the working layer. (**a**) The first stage of preheating; (**b**) the second stage of preheating; and (**c**) the third stage of preheating.

**Figure 5 materials-15-07699-f005:**
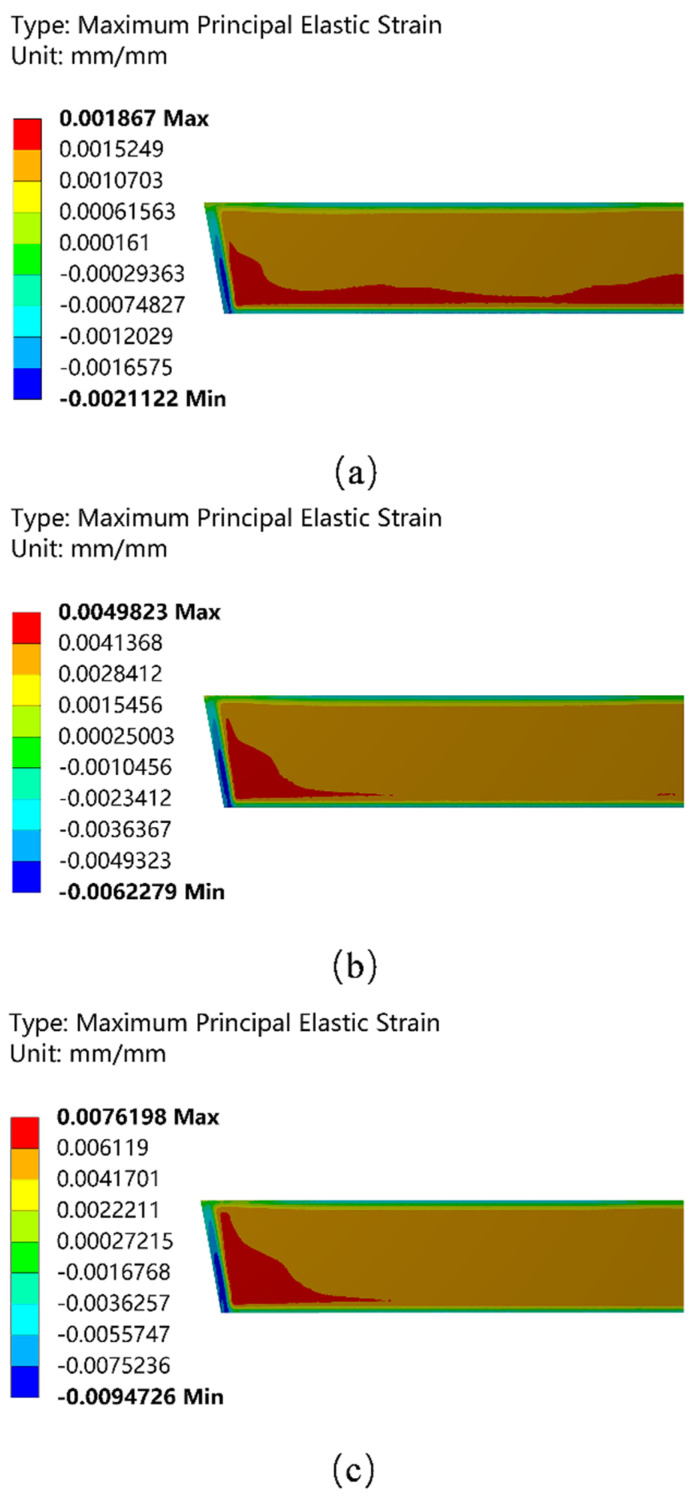
Distribution of the maximum principal elastic strain field of the working layer. (**a**) The first stage of preheating; (**b**) the second stage of preheating; and (**c**) the third stage of preheating.

**Figure 6 materials-15-07699-f006:**
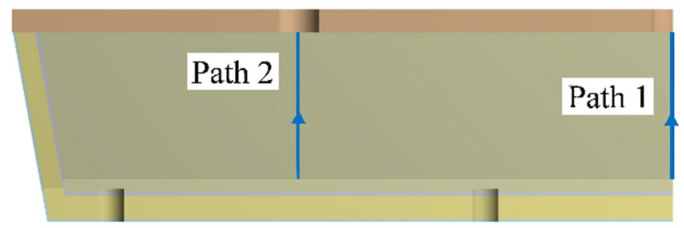
Path 1 and path 2.

**Figure 7 materials-15-07699-f007:**
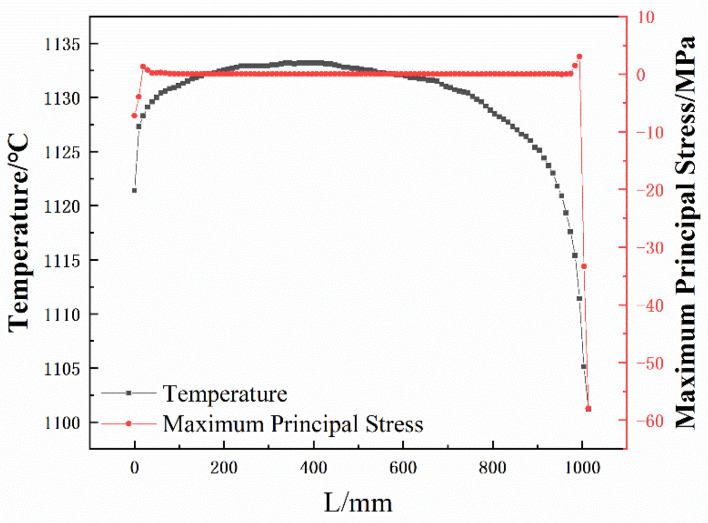
Temperature and maximum principal stress distribution on path 1.

**Figure 8 materials-15-07699-f008:**
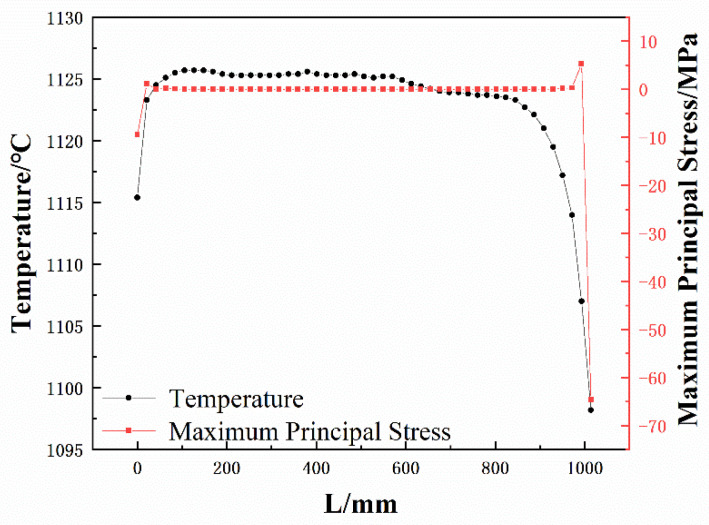
Temperature and maximum principal stress distribution on path 2.

**Figure 9 materials-15-07699-f009:**
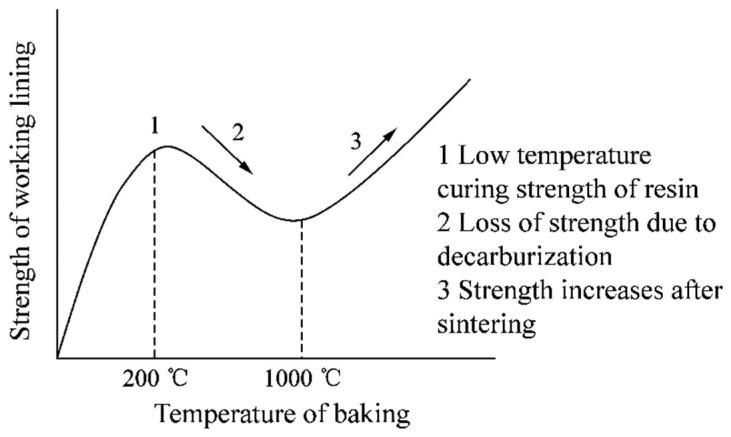
Strength characteristics of the dry vibrating material of the working layer [24].

**Table 1 materials-15-07699-t001:** Physical properties of refractory linings.

	*Ρ* (Kg·m^−3^)	*C_p_* (J·kg^−1^k^−1^)	*k* (w·m^−1^k^−1^)	*α* (°C^−1^)	E (MPa)
Original layer	3330	988.6	-	1.36 × 10^−5^	7500
Decarbonized layer	2349	988.6	-	1.36 × 10^−5^	880
Permanent layer	2600	800	-	6 × 10^−7^	5700
Insulation layer	850	816.4	0.17	5 × 10^−6^	1.2 × 10^5^
Steel shell layer	7820	502	46.4	1.17 × 10^−5^	2 × 10^5^

**Table 2 materials-15-07699-t002:** Thermal conductivity of materials at different temperatures.

Temperatures (°C)	Original Layer (W·m^−1^k^−1^)	Decarbonized Layer (W·m^−1^k^−1^)	Permanent Layer (W·m^−1^k^−1^)
350	0.377	0.455	0.694
600	0.639	0.686	0.898
800	0.953	1.001	1.077
1000	1.015	1.13	1.27

**Table 3 materials-15-07699-t003:** Input parameters and boundary conditions used for CFD simulations.

Parameter	Value
Density	1.7878 kg·m^−3^
Viscosity	1.37 × 10^−5^ kg·m^−1^s^−1^
Heat capacity	840.37 J·kg^−1^K^−1^
Thermal conductivity	0.0145 W·m^−1^k^−1^
Wall	no-slip
Side wall (heat loss coefficient)	15 W·m^−2^
environment temperature	25 °C

**Table 4 materials-15-07699-t004:** The conditions of the burner nozzle.

	Flow Rate (m^3^·h^−1^)	Velocity (m·s^−1^)	Temperature (°C)
The first stage	355	1.3951	400
The second stage	651	2.5583	800
The third stage	1071	4.2088	1150

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
