# Peer review of "Effect of Preheating Temperature on Thermal–Mechanical Properties of Dry Vibrating MgO-Based Material Lining in the Tundish"

_materials, 2022, doi:10.3390/ma15217699_

Round 1
Reviewer 1 Report
The manuscript is very well written; clear, precise, and easy to understand. A very large amount of work was involved in the study, and as far as I can determine, the work is solid. The results are not always new.
Reviewer 2 Report
Journal: Materials (ISSN 1996-1944)
Manuscript ID: materials-1963914
Type: Article
Title: Effect of preheating temperature on thermal-mechanical properties of dry-vibrating MgO-based material lining in the tundish.
Authors: Xiaodong Deng, Jianli Li, Xiao Xie*.
a) Introduction: add more recent refs from the literature survey? And write the objective of the present work carefully.
b) Model description revised Model Description.
c) Eq.(1) revised Eq. 1 and so on.
d) Author should write the equations for the mechanical parameters have been used with ANSYS software for example stress, strain, … etc.
e) For references, choose recent refs. Please, refer to these refs. are very useful for the different measurement characterization
DOI: https://doi.org/10.1088/1742-6596/1795/1/012052
DOI: https://doi.org/10.1088/1742-6596/1795/1/012059
Best Regards
Reviewer 3 Report
Decision:
Minor revision
Comments
The authors reported the Effect of preheating temperature on thermal-mechanical properties of dry-vibrating MgO-based material lining in the tundish. Overall, the work is good and well-presented. However, the authors should address the following points outlined below to improve scientific quality. After the suggested revisions are carefully addressed, this work may be considered for publication
1. Abstract is not clear several mistakes are there for example line 17 Analysis of the areas where the damage to the working layer occurs during the preheating process. The sentence does not make any meaning. The author should recheck the manuscript for grammatical errors.
2. Author should add a table for the abbreviation used in this article
3. In the results and discussion section Figure 3 needs more explanation.
Reviewer 4 Report
Your article is devoted to modeling the process of heating the lining layer of a ladle for steel, but in
the Review part, you write about what additives in the lining material increased its durability. It is worth
considering more studies devoted to solving the problem of lining resistance precisely through process
modeling.
In formulas 1-5, not all variables are defined in the text. Do not see ui, ε, G2, G3, Sk, k.
In paragraph 2.3, when you set the initial conditions for calculations according to the model, why don't
you write or provide a rationale for the conditions taken? After all, you take, for example, 25% of the
depth of the decarburized layer, but why not 50%? It is worth writing about this in this section, as well
as writing about other parameters that you accept for calculations.
The paper has a practical interest, but in terms of methodology, it should be finalized. It is necessary to
more accurately define all the variables used in the formulas for calculations and describe in more detail
based on which such initial data were taken.
Reviewer 5 Report
This paper deals with the ¨ Effect of preheating temperature on thermal-mechanical properties of dry-vibrating MgO-based material lining in the tundish ¨. The manuscript topic is interesting but needs major revision:
1- The writing in English needs significant revision. Many typos and grammatical errors can be detected in the manuscript.
2- The abstract needs more details about the results.
3- The introduction needs to be improved. The benefits and drawbacks of this manuscript are not clear. The authors aimed for what they wanted to present, but the problem they wanted to solve.
4- many pieces of literature are available that are related to this research and give authors more information, and they can be used in the discussion section. Please research and adds more details in the introduction.
5- Avoid bunch citing like ¨[1-5]¨. If the information is valuable add detail from each of them and discusses their results separately.
6- Used Equations need references.
7- It is not clear whether the data of Tables 1 and 2 are original or taken. If they are original, the author should add an experimental section to the manuscript and add details, if the data are taken, then they need a reference.
8- The discussion is weak, please improve it.
9- The main part that was missed in this paper is the validation of model output. How did you validate your simulation output?
Round 2
Reviewer 4 Report
In general, the authors have corrected my comments. I think the work can be published.
Reviewer 5 Report
The authors address all comments. The paper can be published.